# Qualitative documentary analysis of guidance on information provision and consent for the introduction of innovative invasive procedures including surgeries within NHS organisations' policies in England and Wales

Cynthia A Ochieng [1] Hollie Richards [1] Jesmond Zahra [1] Sian Cousins,[1] Daisy Elliott [1] Nicholas Wilson,[1] Sangeetha Paramasivan,[1] Kerry N L Avery [1] Johnny Mathews,[1] Barry G Main [1,2] Robert Hinchliffe,[1,3] Natalie S Blencowe [1,2] Jane M Blazeby [1,2]

For numbered affiliations see end of article.

**Correspondence to**
Dr Cynthia A Ochieng;
c.ochieng@bristol.ac.uk

## ABSTRACT

**Objective** To review guidance, included in written local UK National Health Service (NHS) organisation policies, on information provision and consent for the introduction of new invasive procedures- including surgeries, and devices (IPs/Ds).

**Design** A qualitative documentary analysis of data on patient information provision and consent extracted from policies for the introduction of IP/Ds from NHS organisations in England and Wales.

**Setting** NHS trusts in England and health boards in Wales, UK.

**Participants** Between December 2017 and July 2018, 150 acute trusts in England and 7 health boards in Wales were approached for their policies for the introduction of new IP/Ds. In total, 123 policies were received, 11 did not fit the inclusion criteria and a further policy was included from a trust website resulting in 113 policies included for review.

**Results** From the 113 policies, 22 did not include any statements on informed consent/information provision or lacked guidance on the information to be provided to patients and were hence excluded. Consequently, 91 written local NHS policies were included in the final dataset. Within the guidance obtained, variation existed on disclosure of the procedure's novelty, potential risks, benefits, uncertainties, alternative treatments and surgeon's experience. Few policies stated that clinicians should discuss the existing evidence associated with a procedure. Additionally, while the majority of policies referred to patients needing written information, this was often not mandated and few policies specified the information to be included.

**Conclusions** Nearly a fifth of all the policies lacked guidance on information to be provided to patients. There was variability in the policy documents regarding what patients should be told about innovative procedures. Further research is needed to ascertain the information and level of detail appropriate for patients when considering innovative procedures. A core information set including patients' and clinicians' views is required to address variability around information provision/consent for innovative procedures.

## STRENGTHS AND LIMITATIONS OF THIS STUDY

⇒ This is the first study to systematically review guidance within NHS policies on patient information provision and consent when introducing new invasive procedures and devices in England and Wales.

⇒ The study explores not just the types of information included in the policies but also the level of detail.

⇒ The study cannot ascertain if and how informed consent practices are implemented by clinicians and/or whether other regulatory processes play a role in patient consent.

## INTRODUCTION

A singular definition of surgical innovation is challenging as it can be contextualised to different aspects of a procedure.[1] This may include an entirely novel or modified anatomical approach, differences in the use of devices and technology or changes to a patient group.[2] The degree of modification from standard procedures that constitutes an innovation is also often debated with implications on the governance required for the procedure.[2] Innovation in surgery has transformed how clinical care is delivered; new

surgical techniques and technologies have resulted in improved outcomes, shorter recovery times and fewer adverse events following surgery.[3–5] National guidance for good surgical practice advices clinicians intending to introduce innovative procedures and/or devices (IP/Ds) to contact the interventional procedures programme at the National Institute for Health and Care Excellence (NICE) to find out the novelty status of the procedure or to register it and follow local NHS policies for approval if not being introduced as an NHS Ethics Committee research programme.[6 7] The guidance also states that clinicians should obtain appropriate training in the procedure and participate in ongoing training to develop competence as well as support the training of other clinicians,[6] the guidance however does not specify if the surgeons should report on their experience. Additionally, the guidance states that clinicians should contribute to the evaluation of the IP/Ds by reviewing progress and auditing outcomes and ensure devices are certified and comply with European standards and that patients are informed of the newness of the procedure as well as associated alternatives.[6]

Nonetheless, as documented in recent reports, surgical innovations are sometimes introduced into clinical practice without sufficient evaluation and pose unknown risks to patients.[8 9] Introducing new invasive procedures via research pathways helps ensure clinicians comply with legal, ethical and scientific standards.[10] Despite increasing demand for evidence-based practice, new surgical procedures are commonly delivered prior to appropriate evaluation and without research ethics approval.[11–15] Working in conjunction with NICE, the UK Department of Health and Social Care published guidelines for managing the introduction of new invasive procedures within the NHS.[16 17] These are relevant to NHS local 'new procedures committees/clinical effectiveness committees'.[18] These guidelines included sections on information provision and obtaining consent from patients particularly for procedures introduced outside the remit of research protocols; these were distributed to NHS chief executives via the Health Service Circular (HSC) in 2003.[16]

Patient autonomy is considered a cornerstone of modern healthcare provision,[19] philosophically it is argued that respecting personal autonomy stems from acknowledging that each person has inherent worth and the ability to choose their own destiny.[20] Beauchamp and Childress[20] further explain that for one to exercise personal autonomy, they need to be intentional, done with understanding and be free from controlling influences of others. Respecting this autonomy hence promotes a person's right to hold views and make their own choices as well as supporting their capacity to make these decisions.[20] Clinicians must work in partnership with patients and take their views, goals and circumstances into consideration when establishing best treatment.[21 22] Informed consent can be seen as a process that fosters patient-centred decision-making by respecting their right to self-determination.[23] In the context of surgical innovation,

in terms of ethics, informed consent is complicated by trying to communicate challenging aspects like the inherent uncertainties within the innovation, surgeon's lack of experience, lack of robust evidence for the procedure and cost implications.[24] There is a scarcity of empirical research exploring informed consent in the context of surgical innovation, consequently, little is known about how innovative surgeries are, or should be, discussed with patients.[25] When innovations are introduced as part of a research study, ethical review processes help ensure that patient information is appropriate and includes detail of the potential risks and benefits.[26] However, when procedures are introduced outside the remit of research studies, these safeguards regarding patient consent for innovative surgeries are not always adhered to.[27]

While guidelines on consent practices for new technologies have recently been published,[4] there persists a paucity of research investigating how these are implemented in clinical practice. The Royal College of Surgeons (RCS) guidance state that as part of the consent process for treatment, patients need to be told of their diagnosis and prognosis, options for treatment including no treatment, the purpose, expected benefits and risks of the procedure, likelihood of success and potential further treatment as well as the clinicians that will be involved in the procedure.[28] The General Medical Council (GMC) guidance also add that patients should be told of the uncertainties with the diagnosis and prognosis as well as those associated with the treatment options.[29] The GMC also state that patients need to be told if the treatment is innovative.[29] Both the RCS and GMC guidance state that the patient should be informed by a clinician who is trained and qualified to provide the treatment, and the RCS advises that clinicians should tell patients their experience with the procedure.[28–30] According to the 2003 HSC recommendations, new invasive procedures which have not been evaluated or recommended by NICE should only be granted approval if special consent measures are put in place.[16] The GMC has produced guidance stating that patients should be given information in their preferred format including written, pictures, audio or other media[29] and the RCS state that where possible written information about diagnosis and treatment should be provided to patients.[6] The consent is expected to be documented in written form in the patients notes as well as through written consent forms in the case of invasive procedures.[6 29] This guidance, however, is not specific for innovative procedures being conducted exclusive of a research setting.

Most individual NHS organisations have policies in place regarding the introduction of new IP/Ds;[18 31 32] however, it is unknown what guidance these policies provide about information provision to patients undergoing innovative procedures. This review aimed to investigate information provision and consent measures for the introduction of new invasive procedures and devices described in policy documents within NHS organisations in England (trusts) and Wales (health boards).

## METHODS

### Study design

A comprehensive qualitative documentary review of guidance on information provision and consent within NHS organisations' policies for the introduction of IP/Ds was undertaken. An IP/D was defined as any procedure where access is gained via an incision, natural orifice or percutaneous puncture or involving devices used inside the body.[33] The findings reported here form part of the INTRODUCE Study.[18 34]

### Data collection

The first step in the qualitative documentary analysis as described by May[35] is conceptualising the documents appropriate for the study. In this research, written NHS policies produced by individual hospitals for the oversight of new and modified invasive procedures were the documents of interest. While it is known that such documents frequently exist, to our knowledge they have not been previously studied or collated. While NHS organisations were asked for their policies and guidance documents for innovative, new and modified procedures, we did not give them a definition of what these involved because of the lack of an agreed national definition. Between December 2017 and July 2018, all acute trusts in England (n=150) and health boards in Wales (n=7) were contacted (via email and telephone calls) and requested for their policies relating to the introduction of an innovative IP/D.[18] Of these, 147 responded and 127 stated that they had the requested policies. Of the 127 trusts and health boards, 4 did not share their policies with the research team. Consequently, 123 policies were received of which 11 were excluded from the review for: being device management policies (n=6), application forms only (n=3), committee terms of reference (n=1) and local safety standards for invasive procedures (n=1). Further detail on the eligibility criteria has been published in the main protocol.[18] In addition, one policy was obtained online from the trust website, resulting in 113 policies available for review.

### Data extraction and analysis

The emphasis of the qualitative documentary analysis was initially to discover and subsequently describe[36] the contents of the policies. The next step in the qualitative documentary analysis involved iterative reading of the policies[37] by HR. This can be described as the initial stage of the interpretation and is aimed at subdividing the document into different broad areas of focus.[38 39] For this study, there were varied areas of focus all relating to innovative procedures, this paper reports on the findings of one of these areas—information provision and consent. All text relating to information provision and consent practice for new procedures and devices was extracted verbatim by HR into a data extraction form. The data extraction form was iteratively developed through reviewing policy documents and previous literature[18] and stored on the online surveys portal (formerly Bristol online surveys). Overall, 20% of the policies were independently reviewed

and data extraction was conducted by SC, JB, JM, HR and JZ. Any disagreements among reviewers were discussed and resolved.

Extracted data relating to information provision and consent were then imported into NVivo (V.12) by JZ for review and analysis using methods derived from content analysis[40] and thematic synthesis.[41 42] This stage is often termed the reflective interpretation stage in qualitative documentary analysis.[38 39] This approach allowed for a thematic synthesis to explore the thematic patterns and depth of information while also permitting quantification of their frequencies through content analysis.[36] The extracted data were iteratively read to ensure familiarisation. Following this, coding categories were inductively formulated from reviewing the policies including the HSC document as in a grounded theory approach.[36] These coding categories were collated into a thematic framework. In total, five coding categories were included in the final thematic framework. To ensure this framework fully encapsulated the data, it was piloted on a sample of policies (15%), by JZ and JM independently before being applied to the full dataset. To promote rigour in the data analysis, frequent discussions were held with the team to discuss and refine the initial findings with reference to the raw data. Once the framework had been populated with all the data, it was possible to have a clear overview of each policy's data and analysis results on information provision and consent. This served as case description[38] from each policy, while also providing numerical frequencies of the policies that contained each theme in the framework. The completed framework also permitted the comparative analysis[38] between different policies highlighting similarities and differences among the policies.

### Patient and public involvement (PPI)

The review was conducted under the National Institute for Health Research (NIHR) Bristol and Weston Biomedical Research Centre (BRC) Surgical Innovation theme, which conducts research aimed at improving safety and transparent translation of IPs/Ds to clinical practice. The NIHR Bristol BRC has a dedicated PPI group, where previous surgical patients contribute their views on different projects. In this study, PPI contributors included their views and experiences of the information they received or would have wanted to receive before and after having innovative surgeries. The PPI group was supportive of this study. Our PPI group agreed with the methods applied for the review of the policy documents as well as the resultant findings.

## RESULTS

### Policies included

Of the 113 policies reviewed, 7 did not include any statements on informed consent/information provision. Of the remaining 106, 15 provided only brief statements on informed consent (eg, 'consent must be obtained') and did not provide any guidance on the content of

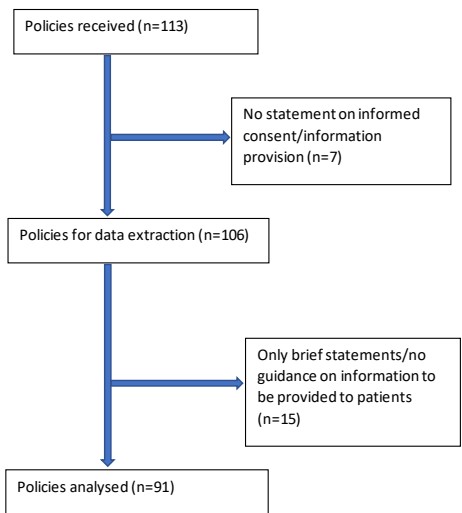

**Figure 1** Flowchart of policy documents reviewed.

information to be provided to patients. This highlights that 19% (22/113) of all the policies did not include any statement on informed consent or lacked guidance on information to be provided to patients. Following discussions within the team, these 15 policies were excluded from further analysis. In total, text from 91 local NHS policies was included in the final dataset (see figure 1).

### Findings
Five categories were included in the final thematic framework and grouped into two main themes: statements about the information clinicians should disclose about a new IP/D being delivered with local NHS policy approval and guidance on the use of written information for this situation. The findings in each of the five categories from the analytic framework are described in detail below. Very few policies included guidance on all of the five categories identified (n=14, see online supplemental appendix 1).

### Statements specifying what information clinicians should disclose prior to delivering an IP/D
#### 'New/special' status of an IP/D
Of the final 91 policies reviewed, 73 (80%) policies stipulated that clinicians should discuss the novelty of a procedure with patients; however, the terminology used within policies varied. In total, 48 (53%) stated the 'special status' (n=44) or 'status' (n=4) of a procedure should be disclosed. This terminology mirrored that used in the HSC document distributed in 2003[16]:

> All patients offered the procedure are made aware of the special status of the procedure.

Of the remaining policies, 20 (22%) used the term 'new' and 5 (5%) included statements that used both 'new' and 'special status'. Notably, only one policy stated clinicians should explain whether the procedure was 'new to the trust' or 'completely new':

An explanation must be given to the patient as to whether the new technique / procedure is either new to the Trust but established elsewhere, or, is a completely new technique / procedure. Policy 81

Three (3%) policies provided guidance on how long clinicians should consider a procedure as new and introduce it to patients as such, however, guidance varied between policies. Specific recommendations were that: (1) standard consent processes should be applied following a successful audit of the procedure, (2) clinicians should review their results and decide when a procedure is an established part of practice and (3) procedures should be introduced as 'new' to the first five patients.

#### Uncertainty, evidence, risks and benefits of a procedure
In total, 58 (64%) policies stated that clinicians should discuss uncertainties surrounding safety and efficacy and/or risks and benefits. Of these, 40 (44%) included an extract from the HSC document[16]:

> Patients need to understand that the procedure's safety and efficacy is uncertain and be informed about the anticipated benefits and possible adverse effects of the procedure.

In total, 12 (13%) of the remaining policies simply stated that patients need to be 'informed' that some uncertainty existed for a procedure.

Five policies provided more comprehensive guidelines for clinicians to follow when discussing uncertainty. For example, one policy stated that the frequency of procedure complications should also be disclosed:

> The most important consideration is that patients (or their parents or carers, when appropriate) should be informed and should understand the risks when offered the procedure. This always means telling them the known risks, but in addition it may mean telling them that there is uncertainty about the frequency (risk) of complications – in particular uncommon and serious ones. Policy 153

Crucially, very few policies (n=3, 3%) stated that clinicians should discuss the evidence or lack thereof associated with a procedure.

#### Alternative treatments
In total, 61 (67%) policies included guidance on discussing alternative treatments, the majority of which (n=45, 49%) used text extracted from the HSC document, which stated that patients need to "be informed about… alternatives, including no treatment".[16] The remaining 16 policies used other synonymous language to communicate the same, for example:

> All alternative options for managing a condition must be described including refusing all treatment. Policy 3

In total, 7 (8%) policies provided further guidance on providing information about alternative treatments. Of

these, 3 (3%) stated that patients should be given the comparative risks and benefits of alternative treatments, 2 (2%) outlined that 'established' treatment options must be discussed with patients and 2 (2%) specified that treatments available elsewhere should also be discussed:

> Patients must be told what alternative treatments are available and what the risks and outcomes are for those treatments, both at this Trust and elsewhere. Policy 134

### Surgeon's experience with the procedure

In total, 57 (63%) policies stated that clinicians should disclose their experience of performing the procedure. Of these, 44 (48%) stated that patients must be made aware of 'the lack of experience of its (IP/D) use'—a statement identical to that used in the HSC document.[16] Of the remaining 13 (14%) policies, 3 (3%) stated that local (ie, within trust) experiences should be discussed, 5 (5%) specifically stated that clinicians should discuss their personal experience with the procedure and 5 (5%) stated both personal and local experience should be discussed. Of the 91 policies, only 4 (4%) stipulated that training related to the procedure and/or any supervision (ie, proctors) should be disclosed to patients:

> Valid consent must include an understanding that the procedure is new, that the practitioner is relatively inexperienced in the technique but (as appropriate) either well supervised or well trained. Policy 128

### Guidance on the use of written information

In total, 67 (74%) policies included guidance regarding the use of written information for patients. Statements, such as the "proposal should include where relevant…a patient information leaflet" and "clinicians should consider providing written patient information", were common throughout many policies.

Although such statements suggested that clinicians should provide written patient information, it was unclear whether this was a mandatory stipulation. Only 30 included explicit instructions that written information must be provided to patients (eg, "there must be written patient information" and "patients must be made aware in writing"). In total, 10 (11%) also specified that written information would be subjected to an internal review before being approved:

> The [local NHS committee] will not consider the new interventional procedure unless an adequate draft PIL accompanies the application. The PIL will need to be approved by the divisional patient information readership panel. Policy 55

Of the 67 policies that referenced written information, only 19 (21%) included guidance on the content that should be included. Of these, the specified contents of the written information (and associated number of policies) were: evidence (n=1), surgeon's experience (n=3), special status/novelty (n=9), NICE status (n=4), alternative treatments (n=5) and potential risks and benefits (n=11), see online supplemental appendix 2.

An example of a more comprehensive statement is provided below, which specified that written information must include the NICE status, potential risks and benefits and alternative treatments to the IP/D. However, no singular policy specified that all the components listed above should be included in the written information.

> The information must include the safety and efficacy of the procedure, whether or not it has been favourably assessed by NICE and the risks, benefits and alternatives of the procedure. Policy 102

## DISCUSSION

This is the first study to investigate guidance included in local written NHS policies on information provision and informed consent for patients being offered new IP/Ds in NHS hospitals (which are not part of a research protocol). Interestingly, nearly a fifth (22/113) of all the policies did not include any statement on informed consent or lacked guidance on information to be provided to patients. Where guidance was provided, there was variation across policies' regarding disclosure of the novel status of a procedure, potential risks and benefits (and uncertainties), alternative treatments and surgeon's experience. We also found that while the majority of policies referred to patients receiving written information, this was often not a mandatory requirement and many policies did not state what information should be included. This highlights the considerable variation in NHS organisations' guidance for information provision and informed consent for innovative IP/Ds. Additionally, many policies had similar wording to the HSC guidance document[16] pointing to the intertextuality often found in bureaucratised documents where texts may appear similar and cross-referencing is not uncommon.[37] It can be postulated that many of the polices we received had been formulated based on the HSC guidance document,[16] any bid to update the policies may require the issuance of a similar guidance through the HSC to NHS organisations.

Written patient information provided prior to surgery is crucial to improving patients' understanding of surgery[43] and is recommended in guidance by RCS[4] as well as NICE.[44] Fundamentally, informed consent should be an exercise of patient autonomy by ensuring that patients understand the procedure and its implications as well as being free of any controlling influences.[20] The General Medical Council (GMC) and NICE further outline that patients should be given information in a format they prefer including written, audio, translated, through pictures or online.[22 44] Our findings demonstrate the absence of a mandatory requirement for specific written information when patients are being offered innovative procedures which could lead to patients lacking appropriate information and affect their ability to exercise informed consent.

The inability to exercise informed consent philosophically contravenes the autonomy of the patient and in so doing impacts on their power and positioning within the clinical decision-making process.[20] Documenting the consent in consent forms before a patient undergoes an invasive procedure is usual practice. The RCS cautions that the consent process should be more than just form filling, their guidance on good surgical practice highlight that it should be informed decision-making supported by clarity in explanation of the diagnosis, prognosis and available procedures.[6] Their guidance also states that the consent forms should be completed at the end of the discussion, stored in the patient notes alongside a description therein of the discussions had and reaffirmed on the day of the procedure.[6] The RCS and GMC guidance, however, provide limited advice on informed consent for innovative procedures. This is the subject of ongoing research at the Bristol BRC.

Regarding the content of the information provided, the GMC asserts that patients must be told the nature of their treatment option, including whether it is an innovative option.[22] While our findings demonstrate that the majority of policies stated that the 'special status' of the procedure needs to be disclosed to patients, it is possible that this disclosure does not explicitly reveal the novelty of the procedure. Additionally, only 3 (3%) policies identified how long the procedures could be considered new or possessing a special status, resulting in ambiguity as to the length of time patients needed to be told about the procedure's newness. Standardising the length of time for which IP/Ds should be considered innovative is unpragmatic because different procedures and devices require different times, data and length of experience to move from the novel status to standard care.[45] According to the IDEAL framework, surgical innovation can be divided into four main stages: innovation, development and exploration, assessment, and long-term implementation and monitoring with each stage requiring its own unique time.[45] The point at which the IP/D is no longer an innovation is therefore more of a conceptual exercise at the long-term implementation stage of the innovation. At that point, the procedure is a part of routine practice and special consent relating to innovative procedures is no longer considered to be needed.[45] In view of these difficulties, it is considered that magnitude of risk and unknown risk is a more important way of defining 'innovation' or something that differs from routine practice.[1] The GMC also states that potential risks, benefits and uncertainties should be discussed with patients.[22] This is also reiterated in the NICE guidelines for shared decision-making, outlining that clinicians should openly discuss the risks, benefits and consequences of treatment choices with patients[44] as has debates in other clinical forums.[24] Although some of the policies examined highlighted the need to discuss risks, benefits and uncertainties with patients, only 3 (3%) stated that clinicians should discuss the evidence associated with the procedure. This was a notable omission across most of the policies given that

discussion of evidence is an element of informed consent identified as important within multiple guidelines.[4 22 46]

Previous research has demonstrated that clinicians find discussing innovation with their patients challenging.[25] The absence of detail within local written NHS policies is likely to exacerbate this issue and highlights the need for more comprehensive and standardised guidelines. Such guidance could help support clinicians in openly discussing innovative surgery with patients. The basis for the amount of detail or depth of information to share with patients has been debated previously, with suggestions including: detail that a typical physician would share, detail that an average patient would understand and bespoke detail for individual patients.[47] In the UK, the expectation is for clinicians to tailor the information to individual patients, taking into consideration the individual patient's values and desires.[48] Previous research has demonstrated that clinicians' opinions on what information is important and how much information to provide to patients can be varied, when offering innovative surgery,[49 50] a phenomenon that can lead to gaps in patient information provision.

It has been postulated that informed consent in reference to surgical innovation (within neurology) needs to include discussions on novelty, risk/benefit ratio, surgeon's experience, potential conflicts of interest, alternatives to the innovative procedure and knowledge or lack thereof of the long-term outcomes.[51] Whether clinicians are required to declare conflict of interest when offering patients innovative procedures is a subject of debate. Guidance from the GMC state that if relevant, clinicians should inform patients of any conflicts of interest that they or their organisation may have.[29] While financial conflicts are expected to be made explicit, there is less guidance about declaring other conflicts of interest.[52] This is pertinent to innovation where personal prestige can be a relevant factor. The ongoing research at the Bristol BRC is exploring these issues with the aim of making recommendations to improve transparency. However, our findings highlight that the policies used across all clinical departments contain variable guidance across NHS organisations, this could lead to uncertainty on what exactly patients need to be told before, during and after undergoing innovative invasive procedures. While our study identified different types of information to share with patients, further research is needed to ascertain which types of information and what level of detail would be appropriate and helpful to patients when considering innovative procedures. Of particular interest would be the development of a core information set which is the minimum information to be given by a clinician in all consultations for a particular operation[53] and in this case within the context of innovative invasive procedures.[53 54]

A major strength of this study is that it has sought to systematically evaluate guidance on patient information and consent in local written NHS policies on introducing new IP/Ds in England and Wales. To date,

very few studies have conducted research on informed consent in the context of surgical innovation, and to our knowledge, none have explored local written NHS policies. Our study highlights the absence of and the variability in guidance, therefore raising concerns about the information patients might receive. Adequate information provision to patients constitutes part of the governance for the introduction of new invasive procedures; this is needed not only to protect patients from ineffective and potentially harmful surgeries but also to uphold their autonomy in the decision-making for treatment which is a key element of biomedical ethics.[20] A limitation of this study is that we cannot ascertain from the current findings if and how informed consent practices are implemented by clinicians, whether they know about these policies and whether other regulatory processes within NHS organisations play a role in patient consent. Additionally, while our findings provide an overview of governance systems used across NHS organisations, it is important to note that a few of the NHS organisations approached did not respond to our request for policy documents (n=9, 5%). Of those that responded, there is the potential that not all relevant documents (such as explanatoryappendices) were shared with us. These documents may have provided further information on informed consent that could have implications on our findings. Based on our findings, we postulate that comprehensive and standardised guidelines on information provision and consent for innovative IP/Ds could facilitate discussions and provide clarity to clinicians regarding the types of information and the level of detail to disclose to patients. This could enhance patients' understanding and facilitate their decision-making based on robust informed consent.

**Author affiliations**
[1]Population Health Sciences, University of Bristol, Bristol, UK
[2]University Hospitals Bristol and Weston NHS Foundation Trust, Bristol, UK
[3]North Bristol NHS Trust, Westbury on Trym, UK

**Acknowledgements** The authors would like to thank Harry Robertson for his contribution to the reviewing and data extraction of 20% of the policy documents for the study.

**Contributors** JMB is the guarantor and developed the idea for the study with support from SC, HR, DE and RH. The methodology was developed by JMB, RH, SC, HR, DE, KNLA, BGM and NSB. Authors HR, SC, JMB, RH, JZ and JM developed the data extraction form. HR extracted free text from policy documents, JZ analysed the data with support from JM, HR and DE. The manuscript was written by CAO and JZ, with critical revisions made by the wider INTRODUCE Study team (JMB, DE, SC, HR, SP, JM, CAO, JZ, KNLA, BGM, NW and NSB). The National Institute for Health Research Bristol and Weston Biomedical Research Centre Surgical Innovation theme is directed by JMB (theme lead) and RH (deputy theme lead).

**Funding** This study was supported by the National Institute for Health Research (NIHR) Biomedical Research Centre at University Hospitals Bristol and Weston NHS Foundation Trust and the University of Bristol, the MRC ConDuCT-II (Collaboration and innovation for Difficult and Complex randomised controlled Trials In Invasive procedures) Hub for Trials Methodology Research (MR/K025643/1) and an NIHR senior investigator award (NF-SI-0514-10114).

**Disclaimer** The lead author (CAO) affirms that the manuscript is an honest, accurate, and transparent account of the study being reported; that no important aspects of the study have been omitted, and that any discrepancies from the study as planned have been explained.

**Competing interests** None declared.

**Patient and public involvement** Patients and/or the public were involved in the design, or conduct, or reporting, or dissemination plans of this research. Refer to the Methods section for further details.

**Patient consent for publication** Not applicable.

**Ethics approval** Not applicable.

**Provenance and peer review** Not commissioned; externally peer reviewed.

**Data availability statement** Data available on request due to privacy/ethical restrictions.

**ORCID iDs**
Cynthia A Ochieng http://orcid.org/0000-0002-5574-6059
Hollie Richards http://orcid.org/0000-0001-9181-7005
Jesmond Zahra http://orcid.org/0000-0002-7947-2216
Daisy Elliott http://orcid.org/0000-0001-8143-9549
Kerry N L Avery http://orcid.org/0000-0001-5477-2418
Barry G Main http://orcid.org/0000-0003-0622-805X
Natalie S Blencowe http://orcid.org/0000-0002-6111-2175
Jane M Blazeby http://orcid.org/0000-0002-3354-3330

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
