## [Reviewer comments · BMJ Open]

ARTICLE DETAILS

TITLE (PROVISIONAL)	A qualitative documentary analysis of guidance on information provision and consent for the introduction of innovative invasive procedures including surgeries within NHS organisation policies in England and Wales.
AUTHORS	Ochieng, Cynthia; Richards, Hollie; Zahra, Jesmond; Cousins, Sian; Elliott, Daisy; Wilson, Nicholas; Paramasivan, Sangeetha; Avery, Kerry; Mathews, Johnny; Main, Barry; Hinchliffe, Robert; Blencowe, Natalie; Blazeby, Jane

VERSION 1 – REVIEW

REVIEWER	Jenkins, Simon University of Warwick
REVIEW RETURNED	11-Jan-2022

GENERAL COMMENTS	I enjoyed reading this manuscript reporting on your study investigating Trust / Health Board-level policies on informing patients about innovative procedures. Please see my suggestions for improvement of the manuscript below. While the findings may be of use to a wider audience than just those interested in surgery, it should be made clearer in the abstract, and possibly in the title, that the focus of this study is on *surgical* interventions. The title should be made more informative with regard to the study type and the approach used. The study methods need to be better defined and described to increase the replicability of the work. It's stated that 'methods derived from' content analysis and thematic synthesis are used, which suggests that the authors used some innovative method, but it is not stated what this is. In general the methods need to be more fully described. It seems that this study could neatly be described as a (qualitative) documentary analysis but this terminology is not used nor is recognition shown of this method or its place in the literature. More detail is needed also on the approach to qualitative analysis. For example, 'To ensure this framework fully encapsulated the data, it was piloted on a sample of policies (15%), by two analysts (JZ and JM) before being independently applied to the full dataset. Data within each theme were reviewed to investigate variations between policy statements on consent.' This suggests that the analysis led to the development of a framework, which was then used to analyse the data further and generate themes. So an iterative approach is suggested but not fully described. I couldn't understand the sentence 'data within each theme were reviewed
--

	to investigate...' so I think more should be done to clarify this and the other aspects of the analysis process. Further to that, the manuscript makes frequent mention of the HSC guidance from 2003. It would be good to know if the thematic analysis derived themes from this guidance as a gold standard (directed analysis?) or if themes were derived independently of this. The authors could include some appraisal of this guidance or comment on whether this is the standard to which they think trusts and health boards should be aiming. Currently it is unclear whether the themes were generated by reference to this standard or exclusively from the data (e.g. a conventional analysis or Grounded Theory approach). Without this description of how themes were generated it is hard to see how the analysis could be repeated. I was surprised to see a STROBE statement included given that until this point in the manuscript, the study was not described as observational. Again this points to more information being required with regard to study methods and design. It would also be an improvement to give context to the manuscript with regard to the philosophical and ethical aspects of information and consent. Some ethical works are cited, but the Beauchamp and Childress reference does not appear until very near the end. More work could be done to show the importance and usefulness of the study's findings by discussing them with reference to the ethical context rather than just existing guidelines like those from NICE, the RCS, and so on. While I do not think it is a barrier to publication, there are some parts where proofreading could improve readability. For example, the phrase 'approvals for the delivery of new invasive procedures, which have not been evaluated or recommended by NICE' is ambiguous; it suggests the approvals should be evaluated, but really you mean the innovations, I think.
--	--

REVIEWER	Fleisher, Lee University of Pennsylvania School of Medicine, Anesthesiology and Critical Care
REVIEW RETURNED	27-Jan-2022

GENERAL COMMENTS	the authors have evaluated the requirements for informed consent for new procedures General comments:  1. Introduction: How would you define a new or innovative procedure? How much different than previous procedures. Within the context of general informed consent, how does this require information on experience with procedure? 2. Method: Please define innovative in the policies. Are there any national standards for informed consent for international readers? 3. Findings: are there national standards for credentialling for new procedures? How much data is required on surgeon experience? 4. Guidance on use of written information: How different is this requirement for written consent? 5. Discussion: Is there any general guidance on written informed consent? Can you discuss the issue of how long something should be considered innovative, new and how similar. Should conflict of interest be a standard?
--

VERSION 1 – AUTHOR RESPONSE

Reviewer: 1

Comments to the Author:

While the findings may be of use to a wider audience than just those interested in surgery, it should be made clearer in the abstract, and possibly in the title, that the focus of this study is on *surgical* interventions. The title should be made more informative with regard to the study type and the approach used.

We have included wording in the title to enhance clarity on the study type, approach and that the focus includes surgeries:

A qualitative documentary review of guidance on information provision and consent for the introduction of innovative invasive procedures including surgeries within NHS organisation policies in England and Wales.

We have also included wording in the abstract to state that the focus is on invasive procedures including surgeries as well as to highlight that this is a documentary analysis:

Objective- To review guidance, included in written local NHS organisation policies, on information provision and consent for the introduction of new invasive procedures- including surgeries and devices (IP/Ds).

Design- A qualitative documentary analysis of data on patient information provision and consent extracted from policies for the introduction of IP/Ds from NHS organisations in England and Wales.

The study methods need to be better defined and described to increase the replicability of the work. It's stated that 'methods derived from' content analysis and thematic synthesis are used, which suggests that the authors used some innovative method, but it is not stated what this is. In general the methods need to be more fully described. It seems that this study could neatly be described as a (qualitative) documentary analysis but this terminology is not used nor is recognition shown of this method or its place in the literature.

Thank you, agreed, this was a qualitative documentary analysis and text has been added in the methods section to clarify this and to explain that it facilitated both a thematic analysis to permit rich description and content analysis to allow for quantification of the findings:

A comprehensive qualitative documentary review of guidance on information provision and consent within NHS organisations' policies for the introduction of IP/Ds was undertaken.

The first step in the qualitative documentary analysis as described by May³² is conceptualising the documents appropriate for the study. In this research written NHS policies produced by individual hospitals for the oversight of new and modified invasive procedures were the documents of interest. Whilst it is known such

documents frequently exist there have to our knowledge not been previously studied or collated.

The emphasis of the qualitative documentary analysis was initially to discover and subsequently describe³³ the contents of the policies. The next step in the qualitative documentary analysis involved iterative reading of the policies³⁴ by HSR. This can be described as the initial stage of the interpretation and is aimed at sub-dividing the document into different broad areas of focus.^{35 36} For this study there were varied areas of focus all relating to innovative procedures, this paper reports on the findings of one of these areas- information provision and consent. All text relating to information provision and consent practice for new procedures and devices was extracted verbatim by HSR into a data extraction form.

More detail is needed also on the approach to qualitative analysis. For example, 'To ensure this framework fully encapsulated the data, it was piloted on a sample of policies (15%), by two analysts (JZ and JM) before being independently applied to the full dataset. Data within each theme were reviewed to investigate variations between policy statements on consent.' This suggests that the analysis led to the development of a framework, which was then used to analyse the data further and generate themes. So an iterative approach is suggested but not fully described. I couldn't understand the sentence 'data within each theme were reviewed to investigate...' so I think more should be done to clarify this and the other aspects of the analysis process.

We have edited the methods section and included further description to make the steps of analysis clearer:

This stage is often termed the reflective interpretation stage in qualitative documentary analysis.^{35 36} This approach allowed for a thematic synthesis to explore the thematic patterns and depth of information while also permitting quantification of their frequencies through content analysis.³³ The extracted data were iteratively read to ensure familiarisation. Following this, coding categories were inductively formulated from reviewing the policies including the HSC document. These coding categories were collated into a thematic framework.

To ensure this framework fully encapsulated the data, it was piloted on a sample of policies (15%), by two analysts (JZ and JM) independently before being applied to the full dataset. To promote rigor in the data analysis, frequent discussions were held with the team to discuss and refine the initial findings with reference to the raw data. Once the framework had been populated with all the data, it was possible to have a clear overview of each policy's data and analysis results on information provision and consent. This served as case description³⁵ from each policy, while also providing numerical frequencies of the policies that contained each theme in the framework. The completed framework also permitted the comparative analysis³⁵ between different policies highlighting similarities and differences among the policies.

Further to that, the manuscript makes frequent mention of the HSC guidance from 2003. It would be good to know if the thematic analysis derived themes from this guidance as a gold standard (directed analysis?) or if themes were derived independently of this. The authors could include some appraisal of this guidance or comment on whether this is the standard to which they think trusts and health

boards should be aiming. Currently it is unclear whether the themes were generated by reference to this standard or exclusively from the data (e.g. a conventional analysis or Grounded Theory approach). Without this description of how themes were generated it is hard to see how the analysis could be repeated.

We apologise for the lack of clarity in our description of the analysis. The analysis was done inductively and codes and themes were formulated from reviewing the policies including the HSC document. This has now been clarified in the methods section.

Following this, coding categories were inductively formulated from reviewing the policies including the HSC document as in a grounded theory approach.³⁵

The discovery that many of the policies had similar wording to the HSC guidance document was a finding from the analysis and points to the intertextuality as added in the discussion section. Rather than proposing the HSC guidance as a standard, we aimed to show that guidance shared through the Health Service Circular seemed to be replicated in the local policies. As described in the findings some wording from the HSC was helpful eg the need to inform patients of uncertainties, however some text could be ambiguous such as the 'special status' of the procedure. Improvements on local policies could therefore be effected through updated guidance disseminated through the HSC:

Additionally, many policies had similar wording to the HSC guidance document pointing to the intertextuality often found in bureaucratized documents where texts may appear similar and cross-referencing is not uncommon.³⁵ It can be postulated that many of the policies we received had been formulated based on the HSC guidance document,¹⁵ any bid to update the policies may require the issuance of a similar guidance through the HSC to NHS organisations.

I was surprised to see a STROBE statement included given that until this point in the manuscript, the study was not described as observational. Again this points to more information being required with regard to study methods and design.

On further reflection, we have decided that the SRQR checklist for reporting is more suited to this study because it is specifically intended to guide the reporting of qualitative studies including qualitative document reviews. A completed SRQR checklist has been uploaded alongside the revised transcript.

It would also be an improvement to give context to the manuscript with regard to the philosophical and ethical aspects of information and consent. Some ethical works are cited, but the Beauchamp and Childress reference does not appear until very near the end. More work could be done to show the importance and usefulness of the study's findings by discussing them with reference to the ethical context rather than just existing guidelines like those from NICE, the RCS, and so on.

Thank you for the suggestion, we have now added text to give context on the philosophical and ethical aspects of informed consent in the introduction:

Patient autonomy is considered a cornerstone of modern healthcare provision,¹⁸ philosophically it is argued that respecting personal autonomy stems from acknowledging that each person has inherent worth and the ability to choose their own destiny.¹⁹ Beauchamp and Childress¹⁹ further explain that for one to exercise personal autonomy, they need to be intentional, done with understanding and be free from controlling influences of others. Respecting this autonomy hence promotes a person's right to hold views and make their own choices as well as supporting their capacity to make these decisions.¹⁹

Informed consent can be seen as a process that fosters patient-centred decision-making by respecting their right to self-determination.²² In the context of surgical innovation, in terms of ethics, informed consent is complicated by trying to communicate challenging aspects like the inherent uncertainties within the innovation, surgeon's lack of experience, lack of robust evidence for the procedure and cost implications.²³

More has also been added in the discussion section:

Fundamentally, informed consent should be an exercise of patient autonomy by ensuring that patients understand the procedure and its implications as well as being free of any controlling influences.¹⁹

The inability to exercise informed consent philosophically contravenes the autonomy of the patient and in so doing impacts on their power and positioning within the clinical decision-making process.¹⁹

This is also reiterated in the NICE guidelines for shared decision making, outlining that clinicians should openly discuss the risks, benefits and consequences of treatment choices with patients³⁶ as has debates in other clinical forums.²³

While I do not think it is a barrier to publication, there are some parts where proofreading could improve readability. For example, the phrase 'approvals for the delivery of new invasive procedures, which have not been evaluated or recommended by NICE' is ambiguous; it suggests the approvals should be evaluated, but really you mean the innovations, I think.

Thank you, we have now edited this to enhance clarity:

According to the 2003 HSC recommendations, new invasive procedures which have not been evaluated or recommended by NICE should only be granted approval if special consent measures are put in place.¹⁵

Reviewer: 2

Comments to the Author:

the authors have evaluated the requirements for informed consent for new procedures

General comments:

1. Introduction: How would you define a new or innovative procedure? How much different than previous procedures. Within the context of general informed consent, how does this require information on experience with procedure?

Thank you, we have included a paragraph discussing the definition of surgical innovation in the introduction:

A singular definition of surgical innovation is challenging as it can be contextualised to different aspects of a procedure¹. This may include an entirely novel or modified anatomical approach, differences in the use of devices and technology or changes to a patient group². The degree of modification from standard procedures that constitutes an innovation is also often debated with implications on the governance required for the procedure².

We have included sentences explaining that while the national guidance state that patients need to be informed by qualified clinicians, the clinicians are not mandated to tell patients about their own experience with the procedure although this is advised by professional bodies:

Both the RCS and GMC guidance state that the patient should be informed by a clinician who is trained and qualified to provide the treatment, and the RCS advises that clinicians should tell patients their experience with the procedure.^{4 24 25}.

2. Method: Please define innovative in the policies. Are there any national standards for informed consent for international readers?

This is an important point and thanks for raising it. One of the aims of the broader part of this study is to examine how innovative is defined within policies. We have found that there are inconsistencies (paper under review BJS). As part of the broader project we examined policies for how innovation was defined. This is reported elsewhere (paper under review BJS). We found inconsistency between policies in the definition of innovation:

Whilst NHS organisations were asked for their policies and guidance documents for innovative, new and modified procedures, we did not give them a definition of what these involved because of the lack of an agreed national definition.

We have now included text describing the national standards for informed consent in the introduction:

The Royal College of Surgeons (RCS) guidance state that as part of the consent process for treatment, patients need to be told of their diagnosis and prognosis, options for treatment including no treatment, the purpose, expected benefits and risks of the procedure, likelihood of success and potential further treatment as well as the clinicians that will be involved in the procedure²³. The General Medical Council (GMC) guidance also add that patients should be told of the uncertainties with the diagnosis and prognosis as well as those associated with the treatment options²⁴. The GMC also state that patients need to be told if the treatment is innovative²⁴

3. Findings: are there national standards for credentialling for new procedures? How much data is required on surgeon experience?

We have now included a description of the national guidance for introducing new procedures. Whilst these exist, however, they are not uniformly adhered to. We have explained that while

surgeons are expected to gain training on the innovation, data relating to surgeon experience was not specifically mandated in it:

National guidance for good surgical practice advises clinicians intending to introduce innovative procedures to contact the national institute for health and Care Excellence (NICE) to find out the novelty status of the procedure or to register it, and follow local NHS policies for approval if not being introduced as an NHS Ethics committee research programme.^{6,7} The guidance also state that clinicians should obtain appropriate training in the procedure and participate in on-going training to develop competence as well as support the training of other clinicians,⁶ the guidance however does not specify if the surgeons should report data on their experience. Additionally, the guidance states that clinicians should contribute to the evaluation of the IP/D by reviewing progress and auditing outcomes and ensure devices are certified and comply with European standards and that patients are informed of the newness of the procedure as well as associated alternatives.⁶

4. Guidance on use of written information: How different is this requirement for written consent?

We have included further description of the guidance on the use of both written information and written consent for patients receiving new IP/Ds without a research study (where a patient information leaflet is mandated). The written information required is encouraged via some policies while the written consent is expected:

The GMC has produced guidance stating that patients should be given information in their preferred format including written, pictures, audio, or other media²⁵ and the RCS state that where possible written information about diagnosis and treatment should be provided to patients.⁶ The consent is expected to be documented in written form in the patients notes as well as through written consent forms in the case of invasive procedures.^{6,25} This guidance, however, is not specific for innovative procedures being conducted exclusive of a research setting.

5. Discussion: Is there any general guidance on written informed consent? Can you discuss the issue of how long something should be considered innovative, new and how similar. Should conflict of interest be a standard?

Thank you for raising these important points. They are controversial and current practice varies, this is why we are conducting this and other studies to improve standards. We have included further description of the guidance on written consent and explained these inconsistencies and need for improvement (including with regard to conflict of interest):

Documenting the consent in consent forms before a patient undergoes an invasive procedure is usual practice The RCS cautions that the consent process should be more than just form filling, their guidance on good surgical practice highlight that it should be informed decision-making supported by clarity in explanation of the diagnosis, prognosis and available procedures⁶. Their guidance also states that the consent forms should be completed at the end of the discussion, and stored in the patient notes alongside a description there-in of the discussions had, and re-affirmed on the day of the procedure⁶. The RCS and GMC guidance, however, provide limited advice on informed consent for innovative procedures. This is the subject of on-going research at the Bristol BioMedical Research Centre.

Whether clinicians are required to declare conflict of interest when offering patients innovative procedures is a subject of debate. Guidance from the GMC state that if relevant, clinicians should inform patients of any conflicts of interest that they or their organisation may have.²⁸ Whilst financial conflicts are expected to be made explicit, there is less guidance about declaring other conflicts of interest.⁵⁰ This is pertinent to innovation where personal prestige can be a relevant factor. The on-going research at the Bristol BioMedical Research Centre is exploring these issues with the aim of making recommendations to improve transparency.

Further discussion has been included highlighting using the IDEAL framework when IP/Ds can no longer be considered new or innovative:

Standardising the length of time for which IP/Ds should be considered innovative is unpragmatic because different procedures and devices require different times, data and length of experience to move from the novel status to standard care³⁴. According to the IDEAL framework, surgical innovation can be divided into four main stages: innovation, development, exploration, assessment and long-term implementation and monitoring with each stage requiring its own unique time.³⁴ The point at which the IP/D is no longer an innovation is therefore more of a conceptual exercise at the long-term implementation stage of the innovation. At that point, the procedure is a part of routine practice and special consent relating to innovative procedures is no longer considered to be needed.³⁴ In view of these difficulties, it is considered that magnitude of risk and unknown risk is a more important way of defining ‘innovation’ or something that differs from routine practice.¹

VERSION 2 – REVIEW

REVIEWER	Jenkins, Simon University of Warwick
REVIEW RETURNED	06-Jul-2022
GENERAL COMMENTS	Thank you for your careful and robust considerations of our suggestions.